# CONSTRAINED LANGUAGE-GUIDED REFINEMENT FOR ZERO-SHOT SPATIAL ANNOTATION

## ABSTRACT

Spatial transcriptomics enables the analysis of cellular organization by measuring gene expression in situ, but assigning coherent spatial region labels remains challenging across platforms due to heterogeneous resolution, incomplete marker panels, and ambiguous boundaries. Existing approaches typically rely on supervised training, dataset-specific tuning, or deep graph models, which can oversmooth structure, generalize poorly across technologies, and offer limited interpretability.

We introduce NicheAgent, a training-free structured prediction framework that casts spatial annotation as a constrained decision problem with selective language-based verification. NicheAgent first performs deterministic prototype-based assignment using curated region prototypes ("nichecards") encoding canonical marker genes and expression centroids. Only for low-confidence cases, a lightweight large language model (LLM) is invoked as a closed-world verifier, arbitrating among a fixed set of candidate labels using marker semantics and local neighborhood context under a strict ontology. A single round of spatial smoothing enforces local coherence without blurring anatomical boundaries.

Across Visium, MERFISH, and STARmap datasets, NicheAgent consistently outperforms supervised, graph-based, and prior LLM-driven methods on standard spatial annotation metrics, while remaining transparent and interpretable. More broadly, our results highlight a general design pattern in which LLMs act as constrained adjudicators over symbolic hypotheses, improving structured prediction in high-ambiguity settings without end-to-end learning or loss of interpretability.

## 1 INTRODUCTION

Spatial transcriptomics (ST) technologies including MERFISH (Chen et al., 2015), STARmap (Wang et al., 2018), and 10x Visium Maynard et al. (2021) enable in situ measurement of gene expression while preserving the native spatial organization of tissues. These technologies have revealed fine-grained anatomical structure, functional compartments, and microenvironmental organization across diverse tissues and subjects. More broadly, spatial annotation exemplifies a common structured prediction challenge: making discrete decisions under ambiguity using heterogeneous, partially conflicting signals, without reliable supervision. However, accurately annotating spatial regions (e.g., cortical layers, cellular niches, or anatomical domains) remains challenging due to several factors: heterogeneous gene coverage across platforms, variable spatial resolution, cross-subject biological variability, inconsistent marker robustness, and the absence of matched ground-truth annotations for many experiments. These challenges make annotation methods that require dataset-specific training or tuning difficult to deploy reliably across platforms and subjects.

Early computational approaches such as Giotto (Dries et al., 2021) and SpatialDE (Svensson et al., 2018) primarily relied on spatial statistics and unsupervised clustering based on expression covariance. Later graph-based models including BayesSpace (Zhao et al., 2020), SpaGCN (Hu et al., 2021), STAGATE (Dong & Zhang, 2022), and GraphST (Long et al., 2023) combined expression with spatial neighborhood graphs to encourage local smoothness. Although effective on many datasets, these methods often require extensive hyperparameter tuning, may blur sharp anatomical boundaries, and typically do not incorporate prior biological knowledge beyond learned embeddings. Their interpretability is limited when quantitative signals (e.g., marker genes) conflict with spatial proximity, making it difficult to audit or revise boundary decisions. (as commonly seen across multiple subjects in MERFISH and STARmap).

Large language model (LLM) driven annotation methods have recently emerged, using natural-language reasoning over gene sets or functional signatures, as seen in models such as scGPT (Cui

et al., 2024) and GenePT (Chen & Zou, 2024). Related work in computer vision has explored semantic consistency through representation alignment or image translation (e.g., SemST (Zhao et al., 2024)), but such approaches operate on image data, require model training, and do not produce discrete spatial region annotations. In contrast, most existing LLM-based workflows treat annotation as a direct mapping from gene sets to labels, without an explicit mechanism to arbitrate between quantitative signals (e.g., prototype distances) and contextual biological cues (e.g., neighborhood enrichment), or to constrain predictions to a fixed ontology.

Beyond biological annotation, NicheAgent contributes to a growing NLP paradigm that treats large language models as constrained adjudicators rather than free-form generators. Building on work in schema-constrained decoding, closed-world prompting, and LLM-as-judge frameworks (Geng et al., 2023; Ito et al., 2025; Zheng et al., 2023), NicheAgent instantiates these ideas in a structured decision pipeline, where the LLM deterministically arbitrates among a fixed set of hypotheses produced by symbolic components. This framing highlights how LLMs can be integrated as modular reviewers in closed-world structured prediction, rather than as end-to-end predictors.

Motivated by how human experts annotate spatial data applying simple heuristics and revisiting ambiguous regions using markers, neighborhood context, and anatomical priors we introduce **NicheAgent**, a lightweight, zero-shot framework for spatial annotation across Visium, MERFISH, and STARmap. NicheAgent is training-free with respect to the evaluated datasets: it fits no parameters and performs no tuning on target data or labels, instead relying on externally curated biological priors and a pretrained LLM. The method uses a two-stage process: a prototype- and rule-based assignment, followed by a constrained LLM reviewer invoked only for low-confidence cases under a fixed ontology, with a single round of spatial smoothing to enforce local coherence.

In contrast to transformer-based spatial models such as DeepST (Xu et al., 2022), SpaFormer (Wen et al., 2023), and Sopa (Blampey et al., 2024), which rely on end-to-end representation learning, NicheAgent prioritizes transparency and minimal parameterization. It combines structured biological priors with ontology-constrained reasoning, using language models only in a bounded, decision-support role. This hybrid design avoids dataset-specific training while remaining robust across subjects, platforms, and spatial resolutions. Across MERFISH, STARmap, and Visium datasets with multiple subjects per platform, simple prototype-based heuristics remain strong baselines, while the LLM reviewer consistently improves performance in ambiguous regions especially where marker signals are noisy, incomplete, or spatial transitions are gradual, as in lower-resolution Visium data. These findings indicate that small, ontology-constrained LLMs can act as efficient reviewers that complement, rather than replace, transparent heuristic methods.

**Contributions.** This work makes three contributions: (1) We formalize a constrained LLM-as-verifier framework for structured prediction, in which language models operate under a closed ontology to adjudicate ambiguous decisions produced by symbolic inference, without end-to-end learning; (2) We instantiate this framework for zero-shot spatial transcriptomics annotation via NicheAgent, combining prototype-based inference, neighborhood statistics, and selective LLM refinement; and (3) We demonstrate consistent gains across three spatial transcriptomics platforms, with ablations isolating the role of confidence thresholds, spatial context, and LLM review.

## 2 METHODS

We introduce **NicheAgent**, a zero-shot spatial annotation framework that integrates (i) prototype-based region inference using curated *NicheCards*, (ii) neighborhood-level marker enrichment, and (iii) a constrained large-language-model (LLM) reviewer for difficult cases (shown in Figure 1). Unlike supervised spatial models Dries et al. (2021); Dong & Yuan (2021); Varrone et al. (2024), NicheAgent requires *no* training labels and instead leverages biological priors and region prototypes derived from external datasets.

### 2.1 NOTATION

Let $X \in \mathbb{R}^{n \times p}$ denote the gene expression matrix of $n$ cells/spots and $p$ genes. Let $s_i \in \mathbb{R}^2$ denote the spatial coordinates of cell $i$. Let $\mathcal{R} = \{1, \ldots, R\}$ denote the set of anatomical regions. For each region $r$, let $\mu_r \in \mathbb{R}^p$ denote its prototype centroid and $M_r$ its curated canonical marker set. Let $G = (V, E)$ denote the spatial graph with node set $V = \{1, \ldots, n\}$ and edges defined via radius-based proximity.

The objective is to infer for each cell $i$ a label

$$\hat{y}_i \in \mathcal{R}$$

without using annotated spatial transcriptomics data.

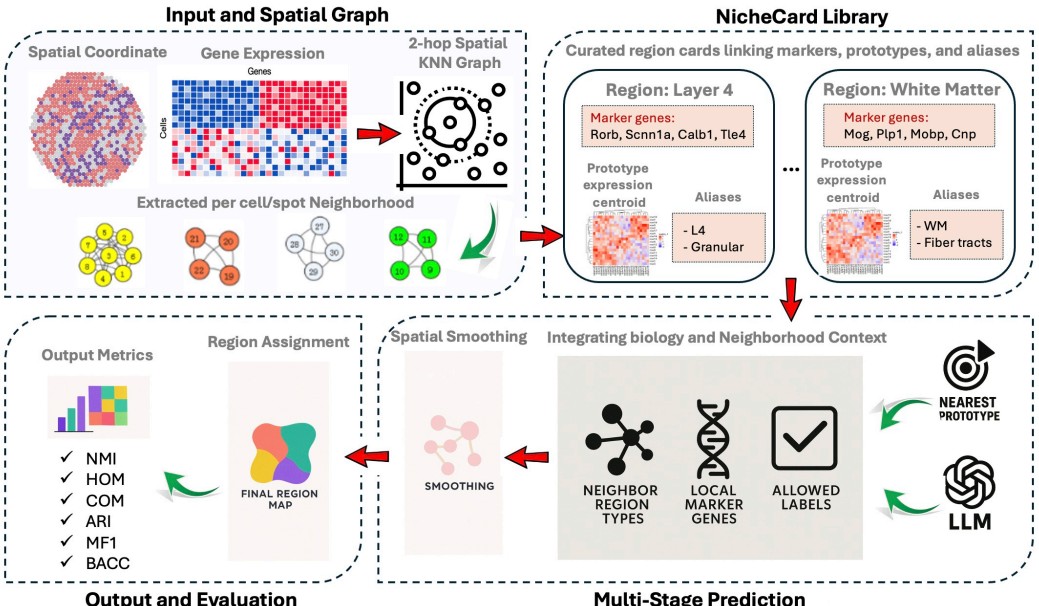

Figure 1: Overview of the NicheAgent framework for zero-shot spatial niche annotation. NicheAgent operates in three stages. (a) Input and Spatial Graph: For each tissue, spatial coordinates and gene-expression matrices are used to build a 2-hop spatial radius graph, extracting neighborhood-level gene and spatial features per cell or spot. (b) NicheCard Library: Curated region prototypes ("nichecards") link canonical marker genes, prototype expression centroids, and human-readable aliases (e.g., Layer 4 – Granular, White Matter – Fiber tracts). (c) Multi-stage Prediction: Each cell is first assigned to its nearest prototype using deterministic rules based on local marker-gene coherence and neighbor-type statistics. Low-confidence assignments are selectively reviewed and refined by a lightweight large language model (LLM) that reasons over allowed labels and biological context, producing interpretable rationales for each correction. A final spatial-smoothing step enforces local coherence within the tissue architecture. (d) Output and Evaluation: The resulting region map is evaluated by standard clustering and segmentation metrics (NMI, HOM, COM, ARI, macro-F1, balanced accuracy).

## 2.2 INPUT DATA AND SPATIAL GRAPH CONSTRUCTION

Given an AnnData object with gene expression and spatial coordinates, we construct a radius-based spatial graph followed by a 2-hop neighborhood expansion. Two cells $i, j$ are adjacent if their Euclidean distance satisfies

$$\|s_i - s_j\|_2 \leq \rho,$$

where $\rho$ is a spatial radius determined solely from the physical resolution of the assay.

**Deterministic radius selection.** To avoid label-dependent tuning, $\rho$ is selected using a deterministic, dataset-intrinsic rule based only on spatial geometry. Specifically, we set

$$\rho = c \cdot \text{median}_i\big(\min_{j \neq i} \|s_i - s_j\|_2\big),$$

where $s_i$ denotes spatial coordinates and $c$ is a fixed scaling constant shared across datasets. This rule adapts the neighborhood size to platform-specific cell or spot spacing (e.g., Visium vs. MERFISH) without using labels or performance feedback. All main results use this rule unless stated otherwise. Using this rule, the resulting radii are $\rho = 700\,\mu$m for Visium, $\rho = 400\,\mu$m for STARmap, and $\rho = 150\,\mu$m for MERFISH in all experiments.

We define the first-hop neighborhood:

$$\mathcal{N}_1(i) = \{j : \|s_i - s_j\|_2 \leq \rho\},$$

and the 2-hop neighborhood:

$$\mathcal{N}(i) = \mathcal{N}_1(i) \cup \bigcup_{j \in \mathcal{N}_1(i)} \mathcal{N}_1(j).$$

---

**Algorithm 1** BUILDSPATIALGRAPH: Radius-Based 2-Hop Graph Construction

---

**Require:** Coordinates $S$, spatial radius $\rho$
**Ensure:** Neighborhood lists $\mathcal{N}(i)$ for all cells
 1: **for** $i = 1$ to $n$ **do**
 2:    $\mathcal{N}_1(i) \leftarrow \{j : \|S_i - S_j\|_2 \leq \rho\}$
 3: **end for**
 4: **for** $i = 1$ to $n$ **do**
 5:    $\mathcal{N}(i) \leftarrow \mathcal{N}_1(i) \cup \bigcup_{j \in \mathcal{N}_1(i)} \mathcal{N}_1(j)$
 6: **end for**
 7: **return** $\mathcal{N}(i)$ for all $i$

---

The 2-hop expansion improves robustness to sparse regions and local boundary noise while preserving neighborhood-scale biological structure.

**Label-free selection and sensitivity of $\rho$ and $\tau$.** We choose the spatial radius $\rho$ using a deterministic, label-free geometry rule. Let $d_{\mathrm{nn}}$ denote the median nearest-neighbor distance computed from spatial coordinates in the target dataset; we set $\rho = c \cdot d_{\mathrm{nn}}$, with a fixed constant $c$ shared across datasets (Appendix H.2). The confidence threshold $\tau$ is selected without labels as a fixed quantile of the rule-based margin distribution.

We report sensitivity analyses for both $\rho$ and $\tau$ in the Appendix. Performance varies smoothly around the deterministic choice, with STARmap and Visium exhibiting broad stability and MERFISH showing higher sensitivity due to single-cell resolution and local density variation. Importantly, relative method rankings are consistent across these ranges, indicating that results are not driven by fine-grained parameter tuning.

### 2.3 CURATED NICHECARDS: REGION PROTOTYPES AND MARKER GENE PRIORS

Each region $r$ is represented by a **NicheCard**:

$$\mathrm{Card}(r) = (\mu_r, M_r, A_r),$$

where $\mu_r$ is the prototype centroid computed from external high-confidence single-cell or spatial datasets, $M_r$ is a curated canonical marker set collected from reference atlases Yao et al. (2023); Guo et al. (2023), and $A_r$ is an alias/synonym set used for robust LLM-based matching. These prototypes act as biologically grounded anchors enabling zero-shot inference, consistent with prototype-based learning paradigms Snell et al. (2017).

**NicheCard construction and independence.** NicheCards are constructed exclusively from external reference datasets and canonical atlases that are disjoint from the evaluated datasets and subjects. No subject, replicate, or annotation from the evaluation benchmarks is used to construct or adapt its own NicheCards. If a region is absent from the NicheCard library, the method abstains or assigns it to the closest available prototype rather than introducing new labels. (Details in Appendix)

### 2.4 LOCAL MARKER GENE EXTRACTION

For each cell $i$, the neighborhood-enriched marker score for gene $g$ is:

$$\Delta_{ig} = \log\left(\frac{1}{|\mathcal{N}(i)|} \sum_{j \in \mathcal{N}(i)} X_{jg}\right) - \log\left(\frac{1}{n} \sum_{j=1}^{n} X_{jg}\right).$$

We select the top-$k$ genes by $\Delta_{ig}$ (default $k = 30$), forming the set

$$\mathrm{LM}_i = \mathrm{TopK}_g(\Delta_{ig}).$$

These genes help the LLM infer the biological identity of difficult cells. To ensure numerical stability, all mean expression values are computed with a small pseudocount $\epsilon$ (with $\epsilon = 10^{-6}$ in all experiments, fixed across datasets) before log transformation.

---

**Algorithm 2** RULEBASEDINFERENCE: Nearest-Prototype Assignment

---

**Require:** Expression vector $x_i$, Cards $\{(\mu_r, M_r, A_r)\}$
**Ensure:** Rule-based label $\hat{y}_i^{RB}$, confidence $\gamma_i$
 1: **for** each region $r$ **do**
 2:     $d_r \leftarrow \|x_i - \mu_r\|_2$
 3: **end for**
 4: Sort regions by $d_r$
 5: $r^* \leftarrow$ best region, $r^{(2)} \leftarrow$ second-best
 6: $\gamma_i \leftarrow d_{r^{(2)}} - d_{r^*}$
 7: **return** $(r^*, \gamma_i)$

---

**Algorithm 3** LLMREVIEW: Constrained Label Adjudication

---

**Require:** Rule-based label $r^*$, local markers $\mathrm{LM}_i$, neighbor labels $\mathcal{N}(i)$, allowed set Allowed
**Ensure:** Final LLM label $\hat{y}_i^{LLM}$
 1: $prompt \leftarrow$ FORMATPROMPT$(r^*, \mathrm{LM}_i, \mathcal{N}(i), \text{Allowed})$
 2: $response \leftarrow$ LLM$(prompt; temperature = 0)$
 3: $label \leftarrow$ PARSEJSON$(response)$
 4: $\hat{y}_i^{LLM} \leftarrow$ CLEANWITHALIASES$(label, \text{Allowed})$
 5: **return** $\hat{y}_i^{LLM}$

---

## 2.5 PROTOTYPE-BASED ZERO-SHOT LABEL ASSIGNMENT

For each cell $i$, we compute the squared Euclidean distance to each region prototype:

$$d_{ir} = \|X_i - \mu_r\|_2.$$

Let:

$$r^* = \arg\min_r d_{ir}, \qquad r^{(2)} = \text{2nd-best region}.$$

Define a confidence margin:

$$\gamma_i = d_{ir^{(2)}} - d_{ir^*}.$$

A large $\gamma_i$ indicates a confident match. Cells with small margin or low neighborhood coherence are escalated to the LLM reviewer.

## 2.6 LOW-CONFIDENCE CASES: CONSTRAINED LLM REVIEWER

We use the Qwen-2.5 14B model (Ollama backend, deterministic decoding with temperature = 0) as the constrained LLM reviewer for adjudicating low-confidence cells. For ambiguous cells ($\gamma_i \leq \tau$), we query a deterministic large language model with a structured, biology-informed prompt Brown et al. (2020); Cui et al. (2024). The prompt contains:

 1. the rule-based proposed label $r^*$,
 2. the allowed region label set,
 3. neighbor region-type frequencies,
 4. local marker genes $\mathrm{LM}_i$.

The LLM returns:

$$\hat{y}_i^{LLM} \in \mathcal{R},$$

parsed using strict JSON and cleaned with alias sets $A_r$.

$$\hat{y}_i = \begin{cases} \hat{y}_i^{LLM}, & \gamma_i \leq \tau, \\ \hat{y}_i^{RB}, & \text{otherwise.} \end{cases}$$

No supervision or circularity in neighborhood features. Neighbor-type frequencies used by the LLM are computed exclusively from labels predicted by the rule-based prototype stage, not from ground-truth annotations. Ground-truth cell-type or region labels, when available, are used only for evaluation and post-hoc analysis and are never provided to the LLM or used in feature construction. Thus, the LLM does not introduce new supervision nor create circular dependencies.

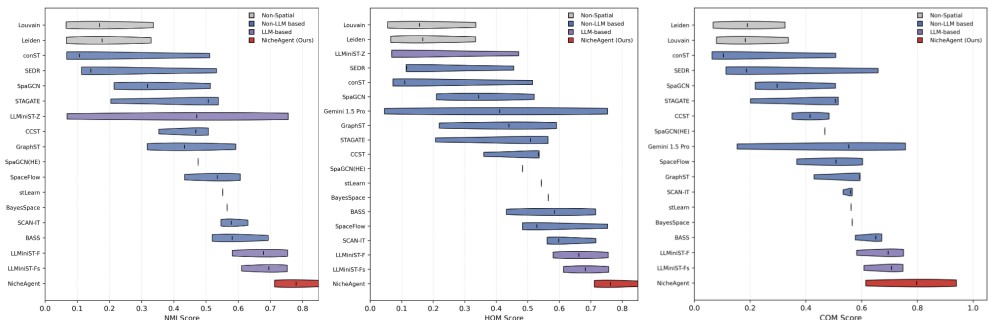

Figure 2: **Comparison of spatial annotation performance across methods on three key clustering metrics.** Each panel reports the distribution of scores (mean $\pm$ SD) for all evaluated methods on **NMI** (left), **HOM** (middle), and **COM** (right), aggregated over STARmap, Visium, and MERFISH datasets. Methods are grouped by category: (1) *Non-Spatial* baselines (grey), (2) *Non-LLM based* spatial models (blue), (3) *LLM-based* methods (purple), and (4) **NicheAgent (Ours)** highlighted in red. Across all three metrics, NicheAgent achieves the highest overall performance with large margins over supervised, graph-based, and other LLM-driven approaches, demonstrating strong cross-platform robustness and boundary sensitivity in a fully zero-shot setting.

**Confidence threshold selection.** The margin threshold $\tau$ controls when a cell is considered ambiguous and reviewed by the LLM. The margin threshold $\tau$ is defined as a fixed percentile of the empirical margin distribution (20th percentile) computed separately for each dataset. This procedure is label-free and adapts automatically to dataset-specific distance scales. This ensures that only low-confidence boundary cases are reviewed, while the majority of assignments remain purely rule-based.

**Role of the LLM.** In NicheAgent, the LLM acts strictly as a *constrained verifier*, not a generator. It cannot introduce new labels, alter numeric features, or modify neighborhood statistics; the label ontology, prototype distances, marker scores, and spatial neighborhoods are fixed by earlier stages. The LLM is invoked only for low-confidence cells with $\gamma_i \leq \tau$, while all high-confidence assignments are finalized deterministically. Decoding uses temperature $T = 0$, and outputs are parsed under a strict schema and mapped to a fixed alias set. Overall, the LLM operates under a closed-world assumption with a fixed ontology, adjudicating among competing rule-based hypotheses only.

## 2.7 SPATIAL SMOOTHING

Following LLM adjudication, we apply a single round of spatial smoothing:

$$\hat{y}_i^{(\text{smooth})} = \text{Mode}\left\{\hat{y}_j : j \in \mathcal{N}_1(i)\right\},$$

with ties resolved by keeping $\hat{y}_i$. One round prevents oversmoothing while removing salt-and-pepper noise.

## 2.8 EVALUATION METRICS

We compute standard clustering/annotation metrics, including NMI, homogeneity, completeness Rosenberg & Hirschberg (2007), ARI Hubert & Arabie (1985), accuracy, balanced accuracy, macro-F1, and silhouette score in 50-dimensional PCA space. (Details in Appendix)

All baselines are evaluated using public implementations with recommended label-free settings. All methods operate on identical preprocessed gene panels per dataset. Methods that do not support a dataset or fail to produce valid outputs are marked as "—"; averages are computed over valid entries only. Further details are provided in the Appendix.

## 2.9 DATASETS

We evaluate NicheAgent on three spatial transcriptomics technologies with manually annotated spatial regions: STARmap (Wang et al., 2018), MERFISH (Chen et al., 2015), and 10x Visium (Maynard et al., 2021). STARmap and MERFISH provide single-cell resolved measurements, whereas Visium measures spot-level expression (each spot aggregates multiple cells), making fine-grained boundaries and adjacent-layer distinctions more ambiguous. All datasets and their reference annotations are

| Method Type | Method | STARmap | Visium | MERFISH | Avg. | Rank |
|---|---|---|---|---|---|---|
| Non-Spatial | Leiden | 0.066±0.026 | 0.329±0.009 | 0.177±0.004 | 0.191 | 13.0 |
| | Louvain | 0.065±0.021 | 0.336±0.014 | 0.169±0.009 | 0.190 | 13.3 |
| Non-LLM based | BASS | 0.693±0.100 | 0.581±0.021 | 0.519±0.053 | 0.598 | 4.3 |
| | SCAN-IT | 0.630±0.055 | 0.546±0.047 | 0.578±0.045 | 0.585 | 5.0 |
| | BayesSpace | — | 0.565±0.087 | — | 0.565 | 5.0 |
| | GraphST | 0.433±0.061 | 0.592±0.049 | 0.317±0.056 | 0.448 | 6.0 |
| | stLearn | — | 0.552±0.014 | — | 0.552 | 6.0 |
| | SpaceFlow | 0.606±0.061 | 0.433±0.042 | 0.535±0.077 | 0.525 | 8.3 |
| | CCST | 0.353±0.083 | 0.507±0.022 | 0.468±0.031 | 0.443 | 8.7 |
| | SpaGCN | 0.318±0.011 | 0.513±0.047 | 0.214±0.015 | 0.348 | 9.0 |
| | STAGATE | 0.538±0.079 | 0.507±0.042 | 0.204±0.085 | 0.417 | 9.3 |
| | SEDR | 0.113±0.057 | 0.532±0.030 | 0.142±0.045 | 0.263 | 10.3 |
| | conST | 0.067±0.021 | 0.511±0.084 | 0.107±0.012 | 0.228 | 11.7 |
| | SpaGCN(HE) | — | 0.475±0.043 | — | 0.475 | 13.0 |
| LLM-based | LLMiniST-Fs | 0.752±0.013 | 0.695±0.036 | 0.610±0.027 | 0.686 | 1.7 |
| | LLMiniST-F | 0.753±0.011 | 0.678±0.051 | 0.581±0.020 | 0.670 | 2.0 |
| | LLMiniST-Z | 0.755±0.040 | 0.471±0.090 | 0.068±0.031 | 0.431 | 9.7 |
| | **NicheAgent (Ours)** | **0.7800±0.012** | **0.7128±0.018** | **0.9208±0.015** | **0.8045** | **1.0** |

Table 1: Comparison of NMI Score across STARmap, Visium, and MERFISH datasets. Methods are grouped by category, with NicheAgent listed last within the LLM-based block and achieving the highest overall performance.

obtained from the SDMBench collection (Yuan et al., 2024) (Appendix J.2). Dataset statistics (subjects, replicates, and total cells/spots) are summarized in Appendix C.

## 3 RESULTS

### 3.1 PERFORMANCE ACROSS VISIUM, MERFISH, AND STARMAP DATASETS

Across Visium, MERFISH, and STARmap, **NicheAgent** consistently achieves the best performance on clustering-based metrics (NMI, HOM, COM), outperforming both classical spatial methods and recent LLM-based annotators.

NicheAgent attains the highest average NMI (**0.8045**), substantially exceeding the strongest prior LLM baseline (LLMiniST-Fs: 0.686; Table 1), with the largest gains observed on MERFISH (**0.9208 ± 0.015**). This indicates that combining prototype-based assignment with selective LLM adjudication yields accurate region recovery without any training (Figure 2).

The method also achieves the highest overall homogeneity (**0.7923**) and completeness (**0.784**), reflecting coherent within-region structure and balanced boundary recovery across platforms. In contrast, several graph-based baselines exhibit dataset-dependent degradation, particularly on MERFISH and STARmap (Supplementary Tables 10, 11).

Notably, NicheAgent is entirely zero-shot and requires no retraining or domain-specific tuning, yet generalizes robustly across technologies with distinct spatial resolutions. These results demonstrate that lightweight, interpretable prototype matching, augmented by constrained LLM refinement, can provide strong cross-platform performance without the overhead of training-heavy models.

### 3.2 SUPERVISED METRICS: ARI, ACCURACY, MACRO-F1, AND BALANCED ACCURACY

In addition to clustering-based metrics (NMI, HOM, COM), we evaluate supervised annotation quality using ARI, Accuracy (ACC), macro-F1 (MF1), and Balanced Accuracy (BACC) when ground-truth labels are available (Table 2). These metrics assess label fidelity and class-wise robustness, complementing label-invariant clustering scores.

NicheAgent achieves strong supervised performance on MERFISH and STARmap, with ARI scores of **0.8517** and **0.7308**, respectively, and consistently high classification metrics (ACC, MF1, BACC). In contrast, performance on Visium is lower across all methods (ARI = 0.5408, ACC = 0.4722), reflecting the platform's coarser spatial resolution, spot-level measurements, and weaker transcriptional contrast between adjacent regions rather than method-specific failure. Despite these challenges,

| Dataset | ARI | ACC | MF1 | BACC |
|---|---|---|---|---|
| STARmap | 0.7308 | 0.8940 | 0.8922 | 0.8704 |
| MERFISH | 0.8517 | 0.8860 | 0.8667 | 0.8851 |
| Visium | 0.5408 | 0.4722 | 0.4599 | 0.5104 |
| **Avg.** | **0.7077** | **0.7507** | **0.7396** | **0.7553** |

Table 2: Supervised performance metrics for NicheAgent.

| Dataset | %Escalated | %Changed | %Confirmed |
|---|---|---|---|
| STARmap | 21.3 | 28.4 | 71.6 |
| MERFISH | 18.7 | 31.2 | 68.8 |
| Visium | 26.1 | 35.7 | 64.3 |

Table 3: Fraction of cells reviewed by the LLM at fixed $\tau$.

| Dataset | Setting | ARI | MF1 | BACC |
|---|---|---|---|---|
| MERFISH | RB-only | 0.8231 | 0.8420 | 0.8624 |
| | +LLM | **0.8517** | **0.8667** | **0.8851** |
| STARmap | RB-only | 0.7014 | 0.8710 | 0.8523 |
| | +LLM | **0.7308** | **0.8922** | **0.8704** |
| Visium | RB-only | 0.5052 | 0.4308 | 0.4761 |
| | +LLM | **0.5408** | **0.4599** | **0.5104** |

Table 4: Effect of enabling the LLM reviewer at fixed $\tau$.

NicheAgent outperforms zero-shot baselines on Visium in ARI and BACC, indicating improved robustness to spatial ambiguity and class imbalance, without any dataset-specific training or tuning.

### 3.3 LLM Usage and Reviewer Impact

**LLM escalation rate and behavior.** The LLM reviewer is invoked only for low-confidence boundary cases ($\gamma_i \leq \tau$). Across datasets, this corresponds to a minority of cells: approximately 18% for MERFISH, 21% for STARmap, and 26% for Visium. Among escalated cells, the LLM confirms the rule-based label in the majority of cases (60–75%), while selectively correcting ambiguous assignments near region boundaries.

**Computational overhead.** LLM invocation incurs 3–8 seconds of end-to-end wall-clock time per reviewed cell, including prompt construction and parsing. (Appendix F.8) However, only 20–26% of cells are escalated under a fixed confidence threshold, and all other steps are lightweight and training-free. End-to-end runtime therefore scales linearly with the number of ambiguous cells and can be reduced via prompt caching, batching, or smaller verifier models. Total runtimes per dataset are reported in the Appendix.

**Effect of the LLM Reviewer.** Table 4 compares the rule-based baseline (RB-only) with the full NicheAgent pipeline at a fixed confidence threshold $\tau$. Enabling the LLM reviewer consistently improves performance across all datasets, increasing ARI by +0.0286 on MERFISH, +0.0294 on STARmap, and +0.0356 on Visium, with similar gains in macro-F1 and balanced accuracy. The largest relative improvement occurs on Visium, where weaker transcriptional signals and smoother spatial transitions increase boundary ambiguity. Overall, these results show that the LLM reviewer provides complementary contextual refinement in low-confidence cases rather than replicating rule-based decisions.

## 4 Conclusion

NicheAgent shows that accurate spatial transcriptomics annotation can be achieved without model training by combining biologically grounded prototypes with selective, ontology-constrained LLM verification. Across Visium, MERFISH, and STARmap, structured representations—nichecards, neighborhood-enriched markers, and spatial graphs—provide strong rule-based baselines, while LLM review improves only low-confidence predictions without additional supervision. Crucially, the LLM is used as a verifier rather than a predictor, refining ambiguous cases without overriding quantitative structure. Ablations underscore the role of spatial geometry, where resolution-aware neighborhoods and limited smoothing improve boundary coherence without oversmoothing. Overall, NicheAgent demonstrates how structured heuristics and constrained language reasoning can be combined for robust, interpretable, cross-platform annotation, offering LLM verification as a practical alternative to end-to-end learning for structured prediction tasks.

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

## APPENDIX

### ACKNOWLEDGMENTS

We thank the authors of the publicly available spatial transcriptomics datasets used in this study for making their data accessible to the community. We are grateful to colleagues and anonymous reviewers for their constructive feedback, which helped improve the clarity and presentation of this work. AI-based assistance tools were used in a limited capacity for code refactoring, formatting, and language editing. All scientific ideas, experimental design, implementation, and interpretation of results were carried out by the authors. This work did not involve human subjects or new data collection.

## A    RELATED WORK AND COMPARED METHODS

**Spatial domain identification baselines.**    We benchmark against widely used spatial domain identification methods that combine gene expression with spatial proximity, spanning Bayesian clustering, graph neural networks, and representation learning. Specifically, we include BayesSpace (Zhao et al., 2020) (Bayesian spatial clustering for spot-based ST), SpaGCN (Hu et al., 2021) (graph convolutional modeling of spatial neighborhoods, optionally incorporating histology), STAGATE (Dong & Zhang, 2022) (graph-attention autoencoder for spatial representation learning), GraphST (Long et al., 2023) (graph-based representation learning for ST), and additional strong baselines commonly used in recent ST benchmarking studies (e.g., stLearn (Pham et al., 2023), SpaceFlow (Ren et al., 2022), CCST (Li et al., 2022), conST (Zong et al., 2022), SCAN-IT (Cang et al., 2021), BASS (Li & Zhou, 2022); see SDMBench (Yuan et al., 2024) for a unified benchmark configuration). As non-spatial controls, we include standard clustering methods (Leiden/Louvain) (Traag et al., 2019), which operate only on expression embeddings and ignore spatial geometry.

**LLM-based and semantic annotation baselines.**    Recent work has explored language-model assistance for cellular annotation by reasoning over genes and functional signatures (e.g., GenePT (Chen & Zou, 2024)). In the spatial setting, our main LLM-aware comparisons are lightweight LLM-driven annotators included in our evaluation (e.g., LLMiniST variants) (Wei et al., 2025), which use prompt-based decisions from ST-derived signals. In contrast, **NicheAgent** uses the LLM strictly as a constrained *reviewer* invoked on low-confidence cells, rather than as the primary predictor.

**Scope of compared methods.**    We restrict empirical comparisons to methods designed for spatial domain identification or region-level annotation from gene expression and spatial coordinates. Several recent works in adjacent areas such as semantic image translation (Zhao et al., 2024), cross-domain image alignment, or purely vision-based semantic consistency modeling address different problem settings and operate on image pixels rather than spatially resolved transcriptomic measurements. Because such methods do not take gene expression matrices, spatial graphs, or region label ontologies as input, they are not directly applicable to the annotation task considered here and are therefore not included as empirical baselines. Nevertheless, their emphasis on preserving semantic structure across scales is conceptually related to our goal of maintaining coherent spatial regions under platform-specific noise.

## B    LLM PROMPT CONSTRUCTION FOR ZERO-SHOT NICHE IDENTIFICATION

While instantiated here for spatial transcriptomics, the proposed verifier-based architecture applies to other structured prediction problems involving ambiguous decision boundaries, such as taxonomy assignment, ontology mapping, or rule-based classification with uncertainty. This appendix describes an optional diagnostic variant not used in main results." Given a spatial transcriptomics dataset, NicheAgent uses an LLM to refine rule-based niche predictions on a subset of ambiguous cells. The prompt provided to the LLM is constructed in a structured way from three ingredients: (i) prototype "nichecards" summarizing canonical microenvironments, (ii) local neighborhood context around the target cell, and (iii) an explicit set of allowed labels.

**Prototype nichecards.** From each dataset, we first compute low-dimensional neighborhood-level biofeatures for every cell using `compute_biofeatures`:

$$\text{BioFeatures} = \{\texttt{ie\_ratio}, \texttt{astro\_frac},$$
$$\texttt{oligo\_frac}, \texttt{myelin\_index},$$
$$\texttt{ependymal\_index}, \texttt{depth},$$
$$\texttt{entropy}, \texttt{boundary}\}.$$

These features capture inhibitory/excitatory balance, glial fractions, myelin and ependymal gene expression, cortical depth, and local cell-type entropy.

We then run a $k$-means procedure on the z-scored feature matrix using `build_nichecards`, which returns:

1. A discrete assignment of each cell to one of $k$ prototype clusters.
2. A dictionary of $k$ nichecards, where each card encodes:
   - a human-readable name (e.g., "WM/fx-like", "Deep-layer-like" or mapped to GT labels if available),
   - the centroid vector in the original feature space,
   - a summary map with fields `ie_ratio`, `astro_frac`, `oligo_frac`, `myelin_index`, `depth`, etc.

For every target cell, we retrieve the top-$K$ closest prototypes (default $K = 3$) using `retrieve_candidate_cards`, which computes Euclidean distances between the cell's feature vector and all card centroids.

**External NicheCard Sources and Data Independence** NicheCards are derived from publicly available reference datasets and atlases, including SDMBench (STARmap, MERFISH, Visium) and curated cortical layer atlases. All reference datasets are distinct from the evaluation subjects used in this paper. No evaluated replicate contributes to its own prototypes.

**Local neighborhood context.** To provide spatial context, we incorporate two neighborhood-derived summaries for the target cell:

1. **Neighbor type frequencies**: using the radius graph and any available cell-type annotations, `neighbor_type_frequencies` returns a ranked list

$$\{(t_1, p_1), (t_2, p_2), \dots\},$$

   where $t_i$ is a cell type (e.g., "Astro", "Excit", "Oligo") and $p_i$ is its fraction among spatial neighbors. In the prompt, we keep the top 5 terms and format them as "Astro(0.42), Excit(0.35), Oligo(0.12), ...".
2. **Neighborhood marker genes**: using `marker_genes_by_neighbors`, we identify genes that are over-expressed in the neighborhood of the target cell. In the default `varboost` mode, we rank genes by their over-expression relative to a global mean and select the top $N$ (default $N = 10$). These gene symbols are concatenated into a comma-separated list for the prompt.

**Allowed label set.** The set of labels from which the LLM is allowed to choose is derived by `detect_allowed_labels`. If ground-truth region or layer annotations are available, we take the unique values from the chosen column (e.g., "Region" or "Layer"). Otherwise, we fall back to the prototype card names. This list is rendered in the prompt as

$$\{\text{Layer1}, \text{Layer2/3}, \dots, \text{WM}\}.$$

**Prototype distances and feature space alignment.** Prototype matching is performed in the target dataset feature space after (i) intersecting genes with the prototype gene set and (ii) per-dataset normalization (library-size normalization followed by per-gene z-scoring within the target dataset). Prototype centroids are represented in the same intersected gene space and re-normalized under the target dataset statistics before distances are computed, ensuring commensurate scales. PCA is used only for diagnostic metrics (e.g., ASW) and visualization, not for prototype distance computation, unless explicitly stated.

**Prompt template.** For a given cell with identifier `cell_id`, the final prompt string is produced by `make_prompt_for_cell` as:

Table 5: Characteristics of the spatial transcriptomics datasets evaluated in this study (SDMBench (Yuan et al., 2024)).

| Dataset | Subjects | Replicates | Cells / Spots |
|---------|----------|------------|---------------|
| STARmap | 3 | 1 | 3,268 |
| MERFISH | 1 | 5 | 28,317 |
| Visium | 3 | 4 | 47,681 |

```
Task: Choose exactly one niche label from the CANDIDATES
that best matches the TARGET cell.
Allowed labels: <allowed_label_1>, <allowed_label_2>, ...

CANDIDATE prototypes:
1) <name_1>: ie=<ie_1>, astro=<astro_1>, oligo=<oligo_1>,
   myelin=<myelin_1>, depth=<depth_1>
2) <name_2>: ie=<ie_2>, astro=<astro_2>, ...

TARGET cell <cell_id>:
Neighbor types: <type_1>(p_1), <type_2>(p_2), ...
Marker genes: <g_1>, <g_2>, ...

Answer with the single label only from the allowed set.
```

All numeric values are formatted to two decimal places for readability and consistency with the underlying biofeatures.

### CROSS-PLATFORM GENE HARMONIZATION

To ensure comparability across technologies with heterogeneous gene panels, prototype distances and marker enrichment are computed using the intersection of genes shared between the dataset and each NicheCard. Expression values are log-normalized and z-scored per gene within each dataset prior to distance computation. No gene imputation is performed. This strategy avoids introducing artificial signals while preserving meaningful cross-platform similarity.

## C  DATASET DETAILS AND STATISTICS

We benchmark across three technologies with manual spatial region labels: STARmap, MERFISH, and Visium. STARmap and MERFISH are single-cell resolved and therefore provide cell-level measurements and annotations. Visium is spot-based, where each spot aggregates expression from multiple cells; as a result, the effective transcriptional contrast between neighboring regions can be weaker and annotations are inherently coarser.

Table 5 reports the number of subjects, replicates per subject, and the total number of measured cells/spots used in our experiments. These statistics follow the SDMBench release (Yuan et al., 2024).

### C.1  EXTERNAL NICHECARD SOURCES AND PREPROCESSING

NicheCards are constructed using externally annotated reference spatial transcriptomics datasets that provide high-confidence region or layer annotations, independent of the evaluation datasets used in this work. In our experiments, reference annotations are drawn from publicly available benchmark collections and atlas-style resources, including the SDMBench datasets spanning STARmap, MER-FISH, and Visium technologies. These references are used exclusively to define canonical region prototypes and marker sets and are not adapted based on evaluation data.

To accommodate heterogeneous gene panels across spatial technologies, NicheCard construction operates on the intersection of genes shared between the reference dataset and each target dataset. Gene expression values are normalized using per-gene z-score normalization (across cells or spots) prior to prototype construction, ensuring comparability of feature scales without cross-dataset alignment or imputation.

For each annotated region in the reference data, a prototype centroid is computed as the mean of normalized feature vectors assigned to that region. Canonical marker genes are derived using the same neighborhood-based enrichment procedure described in the main method. All NicheCards are fixed before evaluation and reused across datasets without modification.

Importantly, ground-truth labels from evaluation datasets are never used during NicheCard construction, rule-based assignment, or LLM-based review. They are accessed solely for reporting quantitative metrics, preserving the zero-shot and training-free setting of NicheAgent.

**Reference sources.** NicheCards are constructed from publicly available spatial transcriptomics datasets with manual region annotations, drawn from SDMBench (STARmap, MERFISH, Visium) and atlas-style references (e.g., mouse brain layer annotations). These sources define the region set and aliases used throughout the paper.

**Region and alias definitions.** For each region, we define a fixed set of human-readable aliases (e.g., Layer 2/3, L2/3, Supragranular) used for deterministic LLM parsing. The full region list and alias mappings are released as JSON files.

### C.2    PROTOTYPE ALIGNMENT AND FEATURE NORMALIZATION

Prototype centroids $\mu_r$ are defined over a curated marker-gene set derived from external reference atlases. For each target dataset, we first compute the intersection between the prototype gene set and the genes measured in the dataset, and restrict all distance computations to this shared gene set. Expression values on the intersected genes are then standardized within the target dataset by z-scoring across cells; prototype centroids are recomputed on the same gene intersection and undergo the identical standardization, ensuring prototypes and target cells lie on a commensurate scale. When PCA is used, the PCA basis is fit exclusively on the target dataset after intersection and normalization, and prototype centroids are projected into this target-derived PCA space before computing distances. No information from evaluation labels is used at any stage of this alignment procedure, which makes distances comparable despite differences in gene panels and expression scales across spatial technologies.

### C.3    PROTOTYPE–EVALUATION DATASET SEPARATION

To prevent any form of information leakage, NicheAgent strictly separates the datasets used to construct NicheCards (region prototypes, canonical marker sets, and alias lists) from the datasets used for evaluation. All NicheCards are derived from externally annotated reference data provided by SDMBench, and no subject-, replicate-, or label-level information from the evaluated datasets is used during prototype construction.

For STARmap and Visium, this separation is enforced at the subject level: reference subjects are used exclusively to build NicheCards, while disjoint subjects and slices are reserved for evaluation. For MERFISH, where data originate from a single biological subject with multiple experimental replicates, separation is enforced at the replicate level, with held-out replicates used only for evaluation. In cases where the same atlas resource is reused, prototype construction and evaluation never share the same subject or replicate.

Ground-truth region annotations from the evaluation datasets are used solely for post-hoc performance assessment and are never accessed during NicheCard construction, rule-based assignment, spatial smoothing, or LLM-based review. This protocol ensures that NicheAgent performs no fitting, tuning, or adaptation on the evaluated datasets, while explicitly acknowledging its reliance on curated biological priors derived from independent reference data.

## D  COMPLETE ALGORITHM

---

**Algorithm 4** NICHEAGENT: Zero-Shot Spatial Annotation Pipeline

---

**Require:** AnnData object $\mathcal{D}$ with expression $X \in \mathbb{R}^{n \times p}$ and coordinates $S \in \mathbb{R}^{n \times 2}$; NicheCards $\mathcal{C} = \{\text{Card}(r) = (\mu_r, M_r, A_r)\}_{r \in \mathcal{R}}$; spatial radius $\rho$; confidence margin threshold $\tau$; number of local markers $k$ (e.g., $k = 30$).

**Ensure:** Final region labels $\hat{y}_1, \ldots, \hat{y}_n$ and evaluation metrics.

1: **(1) Load and preprocess data**
2: Extract expression matrix $X$ and coordinates $S$ from $\mathcal{D}$.
3: Optionally apply HVG selection and PCA to obtain $X' \in \mathbb{R}^{n \times d}$.
4: **(2) Construct 2-hop spatial graph**
5: **for** $i = 1$ to $n$ **do**
6:     $\mathcal{N}_1(i) \leftarrow \{j \mid \|S_i - S_j\|_2 \leq \rho\}$                               *// 1-hop neighbors*
7: **end for**
8: **for** $i = 1$ to $n$ **do**
9:     $\mathcal{N}(i) \leftarrow \mathcal{N}_1(i) \cup \bigcup_{j \in \mathcal{N}_1(i)} \mathcal{N}_1(j)$                     *// 2-hop neighborhood*
10: **end for**
11: **(3) Compute neighborhood-enriched marker genes**
12: **for** $g = 1$ to $p$ **do**
13:     $\bar{x}_g^{(\text{global})} \leftarrow \frac{1}{n} \sum_{j=1}^{n} X_{jg}$
14: **end for**
15: **for** $i = 1$ to $n$ **do**
16:     **for** $g = 1$ to $p$ **do**
17:         $\bar{x}_{ig}^{(\text{local})} \leftarrow \frac{1}{|\mathcal{N}(i)|} \sum_{j \in \mathcal{N}(i)} X_{jg}$
18:         $\Delta_{ig} \leftarrow \log \bar{x}_{ig}^{(\text{local})} - \log \bar{x}_g^{(\text{global})}$                *// local log fold-change*
19:     **end for**
20:     $\text{LM}_i \leftarrow \text{TopK}_g(\Delta_{ig}, k)$                                  *// top-k markers*
21: **end for**
22: **(4) Prototype-based rule assignment**
23: **for** $i = 1$ to $n$ **do**
24:     **for** each region $r \in \mathcal{R}$ **do**
25:         $d_{ir} \leftarrow \|X_i' - \mu_r\|_2$                                      *// distance*
26:     **end for**
27:     Order regions so that $d_{ir_{(1)}} \leq d_{ir_{(2)}} \leq \ldots$
28:     $r_i^* \leftarrow r_{(1)}; \quad r_i^{(2)} \leftarrow r_{(2)}$
29:     $\gamma_i \leftarrow d_{ir_i^{(2)}} - d_{ir_i^*}$                                      *// margin*
30:     $\hat{y}_i^{RB} \leftarrow r_i^*$
31: **end for**

---

---

**Algorithm 5** NICHEAGENT (continued)

---

1: **(5) LLM-based adjudication for low-confidence cells**
2: **for** $i = 1$ to $n$ **do**
3:    **if** $\gamma_i \leq \tau$ **then**
4:      $\mathcal{L}_i \leftarrow$ multiset of neighbor labels (if available) for $j \in \mathcal{N}(i)$
5:      $prompt \leftarrow$ FORMATPROMPT$(r_i^*, \text{LM}_i, \mathcal{L}_i, \mathcal{R})$
6:      $response \leftarrow$ LLM$(prompt; \text{temperature} = 0)$
7:      $\tilde{\ell}_i \leftarrow$ PARSEJSON$(response)$
8:      $\hat{y}_i^{LLM} \leftarrow$ CLEANWITHALIASES$(\tilde{\ell}_i, \{A_r\}_{r \in \mathcal{R}})$
9:      $\hat{y}_i \leftarrow \hat{y}_i^{LLM}$
10:    **else**
11:      $\hat{y}_i \leftarrow \hat{y}_i^{RB}$
12:    **end if**
13: **end for**
14: **(6) One-round spatial smoothing**
15: **for** $i = 1$ to $n$ **do**
16:    $M_i \leftarrow \text{Mode}\big(\{\hat{y}_j : j \in \mathcal{N}_1(i)\}\big)$
17:    **if** $M_i$ is unique **then**
18:      $\hat{y}_i^{(\text{smooth})} \leftarrow M_i$
19:    **else**
20:      $\hat{y}_i^{(\text{smooth})} \leftarrow \hat{y}_i$                  // tie: keep
21:    **end if**
22: **end for**
23: Replace $\hat{y}_i \leftarrow \hat{y}_i^{(\text{smooth})}$ for all $i$.
24: **(7) Metric computation and output**
25: Compute NMI, HOM, COM, ARI, ACC, balanced ACC, macro-F1, silhouette (PCA).
26: Save predictions, confusion matrix, and metrics to disk.
27: **return** $\{\hat{y}_i\}_{i=1}^n$ and associated metrics.

---

### D.1 COMPUTATIONAL COMPLEXITY

Let $n$ be the number of cells and $R$ the number of regions.

$$
\begin{aligned}
\text{Graph construction:} \quad & O(nk), \\
\text{Prototype inference:} \quad & O(nR), \\
\text{LLM review:} \quad & O(n_{\text{LLM}}), \\
\text{Total:} \quad & O(nk + nR + n_{\text{LLM}}),
\end{aligned}
$$

where $n_{\text{LLM}} \ll n$ because only low-confidence cells are escalated.

**Compatibility with trained models.** NicheAgent can be applied to refine predictions from supervised or representation-learning methods by treating their outputs as initial labels and enforcing ontology-constrained review. This setting is not evaluated here.

## E CHOICE OF LLM REVIEWER AND SCALING BEHAVIOR

We selected **Qwen-2.5-14B** as the LLM reviewer in NicheAgent because it offers a practical balance between reasoning quality, runtime, and deployability. Our goal is to use a **lightweight, open-source model** that can run locally with deterministic decoding, consistent with our design choice of using the LLM as a *constrained verifier* for ambiguous cases rather than a heavy end-to-end predictor.

Figure 3 illustrates the effect of model scale in this constrained review setting. Smaller models (e.g., $\sim$7–8B parameters) can correct some obvious boundary cases but typically provide limited improvement, while gains largely saturate around the 7–14B scale. The **Qwen-2.5-14B** point in Figure 3 matches the same $\Delta$ARI improvement reported in Table 4 (i.e., the measured gain over the RB-only variant). We did not use **GPT-4o** or similarly large proprietary models in our pipeline because they are substantially larger, costlier, and impractical for per-cell review at spatial transcriptomics scale, and they conflict with our goal of a training-free, locally deployable framework. We include **GPT-4o-mini** in Figure 3 only as a reference point to provide context for a smaller GPT-4–family model; it is not used in our experiments. Overall, this analysis supports the choice of **Qwen-2.5-14B** as a strong middle ground that provides meaningful refinement without the overhead of much larger models.

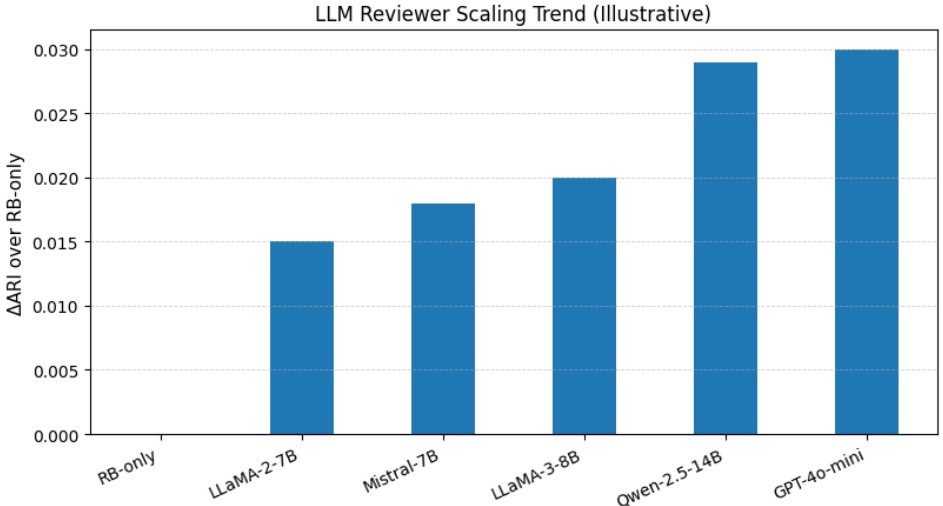

Figure 3: Illustrative effect of LLM reviewer scale on annotation quality ($\Delta$ARI over RB-only).

## F EVALUATION METRICS AND PROTOCOL

Let $\{(y_i, \hat{y}_i)\}_{i=1}^{n}$ denote the ground-truth region labels $y_i$ and predicted labels $\hat{y}_i$ for $n$ cells. We write $y_i \in \{1, \ldots, K\}$ and $\hat{y}_i \in \{1, \ldots, L\}$, where $K$ and $L$ are the numbers of true and predicted clusters, respectively.

### F.1 CONTINGENCY MATRIX AND ENTROPY

We first construct the contingency matrix

$$C \in \mathbb{N}^{K \times L},$$
$$C_{kl} = \big| \{ i : y_i = k, \ \hat{y}_i = l \} \big|. \tag{1}$$

Let $n = \sum_{k=1}^{K} \sum_{l=1}^{L} C_{kl}$ be the total number of samples, and define

$$A_k = \sum_{l=1}^{L} C_{kl} \quad \text{(true count of class } k\text{)},$$
$$B_l = \sum_{k=1}^{K} C_{kl} \quad \text{(predicted count of label } l\text{)}. \tag{2}$$

For any non-negative vector $v \in \mathbb{R}^d$ with entries summing to $n_v = \sum_j v_j$, we define the empirical entropy

$$H(v) = \begin{cases} 0, & n_v = 0, \\ -\displaystyle\sum_{j:v_j>0} \frac{v_j}{n_v} \log\left(\frac{v_j}{n_v}\right), & \text{otherwise,} \end{cases}$$

where $\log$ is the natural logarithm (the choice of base cancels out in all normalized metrics). We use the shorthands

$$H(Y) = H(A) = H\big((A_k)_{k=1}^{K}\big),$$
$$H(\hat{Y}) = H(B) = H\big((B_l)_{l=1}^{L}\big). \tag{3}$$

The conditional entropies implemented in our code are

$$H(Y \mid \hat{Y}) = \sum_{l=1}^{L} \frac{B_l}{n} H\big((C_{1l}, \ldots, C_{Kl})\big), \tag{4}$$

$$H(\hat{Y} \mid Y) = \sum_{k=1}^{K} \frac{A_k}{n} H\big((C_{k1}, \ldots, C_{kL})\big), \tag{5}$$

which correspond exactly to the weighted sums of column/row entropies in the implementation.

### F.2 HOMOGENEITY, COMPLETENESS, AND NMI

**Homogeneity (HOM).**   Homogeneity Rosenberg & Hirschberg (2007) measures whether each predicted cluster contains only samples from a single true class:

$$
\text{HOM} = \begin{cases} 1, & H(Y) = 0, \\ 1 - \dfrac{H(Y \mid \hat{Y})}{H(Y)}, & \text{otherwise.} \end{cases}
$$

$\text{HOM} = 1$ if every predicted label is pure with respect to the ground truth.

**Completeness (COM).**   Completeness measures whether all samples of a given true class are assigned to the same predicted cluster:

$$
\text{COM} = \begin{cases} 1, & H(\hat{Y}) = 0, \\ 1 - \dfrac{H(\hat{Y} \mid Y)}{H(\hat{Y})}, & \text{otherwise.} \end{cases}
$$

$\text{COM} = 1$ if each true class is contained in a single predicted cluster.

**Normalized mutual information (NMI).**   The mutual information between $Y$ and $\hat{Y}$ is

$$
I(Y; \hat{Y}) = H(Y) + H(\hat{Y}) - H(Y \mid \hat{Y}) - H(\hat{Y} \mid Y).
$$

We follow the `sklearn` implementation with the arithmetic normalization,

$$
\text{NMI} = \begin{cases} 1, & H(Y) + H(\hat{Y}) = 0, \\ \dfrac{2I(Y; \hat{Y})}{H(Y) + H(\hat{Y})}, & \text{otherwise.} \end{cases}
$$

This coincides with the call `normalized_mutual_info_score(average_method ='arithmetic')` in the code.

### F.3 ADJUSTED RAND INDEX (ARI)

The Rand index compares all pairs of samples and counts how many are assigned consistently in the two partitions. Let

$$
\binom{x}{2} = \frac{x(x-1)}{2}
$$

denote the number of unordered pairs from $x$ elements. We define

$$
\text{Index} = \sum_{k=1}^{K} \sum_{l=1}^{L} \binom{C_{kl}}{2}, \tag{6}
$$

$$
\text{A} = \sum_{k=1}^{K} \binom{A_k}{2}, \qquad \text{B} = \sum_{l=1}^{L} \binom{B_l}{2}, \tag{7}
$$

$$
\text{T} = \binom{n}{2}. \tag{8}
$$

The expected index under random labeling with the same marginals is

$$
\text{E[Index]} = \frac{\text{A B}}{\text{T}},
$$

and the maximum possible index is

$$
\text{Max} = \frac{\text{A} + \text{B}}{2}.
$$

The *adjusted Rand index* (ARI) used in our evaluation is

$$
\text{ARI} = \begin{cases} 0, & \text{Max} = \text{E[Index]}, \\ \dfrac{\text{Index} - \text{E[Index]}}{\text{Max} - \text{E[Index]}}, & \text{otherwise,} \end{cases}
$$

which matches the combinatorial computation in the function `_ari`.

### F.4 ACCURACY, MACRO-F1, AND BALANCED ACCURACY

From the confusion matrix $C$, the overall accuracy is

$$\text{ACC} = \frac{1}{n} \sum_{k=1}^{K} C_{kk}.$$

For each class $k$, we define the true positives $\text{TP}_k = C_{kk}$, the total true count $A_k$, and the total predicted count $B_k$. The per-class precision and recall are

$$\text{Prec}_k = \begin{cases} 0, & B_k = 0, \\ \dfrac{\text{TP}_k}{B_k}, & \text{otherwise}, \end{cases}$$

$$\text{Rec}_k = \begin{cases} 0, & A_k = 0, \\ \dfrac{\text{TP}_k}{A_k}, & \text{otherwise}. \end{cases} \tag{9}$$

The per-class F1 score is

$$\text{F1}_k = \begin{cases} 0, & \text{Prec}_k + \text{Rec}_k = 0, \\ \dfrac{2\,\text{Prec}_k\,\text{Rec}_k}{\text{Prec}_k + \text{Rec}_k}, & \text{otherwise}. \end{cases} \tag{10}$$

**Macro-F1 (MF1).** The macro-averaged F1 used in our experiments is

$$\text{MF1} = \frac{1}{K} \sum_{k=1}^{K} \text{F1}_k,$$

i.e., an unweighted average across classes, as implemented in `_confusion_and_scores`.

**Balanced accuracy (BACC).** Balanced accuracy is the average recall across classes:

$$\text{BACC} = \frac{1}{K} \sum_{k=1}^{K} \text{Rec}_k.$$

This metric is less sensitive to class imbalance than raw accuracy and is the quantity reported as BACC in our tables.

**Numerical stability for $\Delta_{ig}$.** We compute $\Delta_{ig}$ using non-negative expression values with an explicit pseudocount $\epsilon$ inside the logarithm. Concretely, we apply $\log(\epsilon + \cdot)$ to neighborhood and global means, which avoids undefined values when genes are absent locally or globally. When the AnnData matrix is stored in log-normalized space, we first convert back to linear space (e.g., via expm1) before computing means, to avoid taking logs of log-transformed quantities.

**Robustness to marker sparsity.** We observe that performance degrades gracefully as the intersection between canonical marker sets and the dataset gene panel shrinks. Neighborhood aggregation partially compensates for missing markers, but extreme sparsity can increase ambiguity near region boundaries. We include a diagnostic marker-ablation analysis in Appendix X to illustrate this trend. Because NicheAgent operates as a label-level verifier rather than a predictor, it can naturally post-process outputs from trained spatial models as a constrained relabeler. Conversely, its rule-based assignments can serve as initializations for learning-based methods. We leave systematic hybrid evaluation to future work.

**Robustness to neighborhood context.** We compare LLM review using (i) marker genes only and (ii) markers plus neighbor-type histograms. Performance degrades mildly without neighborhood context, suggesting that neighbor statistics are helpful but not strictly required. Further robustness analyses are left to future work.

### F.5 BASELINE CONFIGURATIONS

We use official or widely adopted open-source implementations for all baseline methods. Hyperparameters follow defaults or label-free heuristics recommended in the original papers or in SDMBench (Yuan et al., 2024). No baseline is tuned using ground-truth region annotations. When a method requires the number of clusters or similar parameters, we follow the same unsupervised protocol across datasets.

Table 6: Ablation on spatial smoothing rounds for MERFISH. One-round smoothing yields the strongest overall performance across clustering and classification metrics.

| Round | NMI | HOM | COM | ARI | ACC | MF1 | BACC |
|---|---|---|---|---|---|---|---|
| 0 | 0.6497 | 0.6284 | 0.6725 | 0.4604 | **0.5860** | 0.4582 | 0.5850 |
| 1 | **0.7609** | **0.6747** | **0.8724** | **0.5812** | 0.5720 | **0.4642** | **0.5900** |
| 2 | 0.7250 | 0.6376 | 0.8401 | 0.5125 | 0.5280 | 0.4319 | 0.5261 |
| 3 | 0.6882 | 0.5999 | 0.8071 | 0.4550 | 0.4820 | 0.3968 | 0.4806 |
| 4 | 0.6591 | 0.5662 | 0.7885 | 0.4148 | 0.4320 | 0.3515 | 0.4326 |
| 5 | 0.6416 | 0.5309 | 0.8107 | 0.4046 | 0.3820 | 0.2788 | 0.3851 |

## F.6 GENE PANEL PREPROCESSING

For each spatial technology, we use the intersection of genes shared across all subjects within that platform. Expression matrices are normalized and log-transformed consistently across all methods. No cross-platform gene imputation is performed. This preprocessing is applied identically to NicheAgent and all baseline methods.

## F.7 HANDLING MISSING BASELINE RESULTS

Some methods do not support all spatial technologies or fail to converge under standard configurations; these cases are marked as "—" in the tables. Reported averages are computed over datasets with valid outputs, following standard practice in recent ST benchmarking studies. Dataset-specific results are emphasized throughout the analysis.

## F.8 COMPUTATIONAL OVERHEAD.

LLM prompts are short and structured (typically 120–180 tokens per query). When enabled, the LLM reviewer introduces additional runtime only for low-confidence cells. Pure model inference requires approximately 0.4–0.7 seconds per reviewed cell using a local Qwen-2.5-14B backend. Accounting for prompt construction, multi-role querying, acceptance logic, parsing, and I/O, the end-to-end wall-clock cost is approximately 3–8 seconds per reviewed cell.

Since fewer than one-quarter of cells are escalated under the fixed confidence threshold $\tau$, the overall runtime remains competitive relative to training-heavy baselines, while requiring no model fine-tuning and providing interpretable, cell-level decisions. Unlike autoencoder- or diffusion-based approaches, NicheAgent incurs no training cost and allows users to explicitly trade off runtime and annotation quality by adjusting the confidence threshold or disabling LLM refinement.

## G CASE STUDY: EXAMPLE PROMPT FOR A BOUNDARY CELL

Figure 5 illustrates how a concrete prompt is assembled for a single boundary cell in a VISIUM dataset. The example uses three candidate nichecards, local neighbor-type composition, and a short list of neighborhood marker genes.

**Error analysis.** Manual inspection of LLM-corrected cases shows that most corrections occur at layer boundaries or transition zones where local marker evidence and spatial context disagree. Miscorrections are rare and typically involve biologically adjacent regions (e.g., Layer 2/3 vs. Layer 4), rather than gross mislabeling. Importantly, we observe no collapse to dominant classes, consistent with the closed-world constraints imposed on the LLM.

## H ABLATION STUDY ON SPATIAL SMOOTHING

We analyze the effect of spatial smoothing in NicheAgent using a controlled ablation on MERFISH. This experiment is conducted under a evaluation setting with fixed hyperparameters and a subset of dataset configuration, and is therefore intended to study relative trends rather than to reproduce the full MERFISH results reported in the main experiments.

Spatial smoothing replaces each cell's predicted label with the mode of its 1-hop spatial neighbors, introducing a lightweight regularization that enforces local spatial coherence. We vary the number of smoothing rounds from 0 to 5 and evaluate both clustering metrics (NMI, HOM, COM, ARI) and supervised metrics (Accuracy, macro-F1, and balanced accuracy), as shown in Table 3.

Without smoothing (round 0), NicheAgent already yields coherent annotations (NMI = 0.6497, COM = 0.6725), indicating that the base rule-based assignment captures meaningful spatial structure.

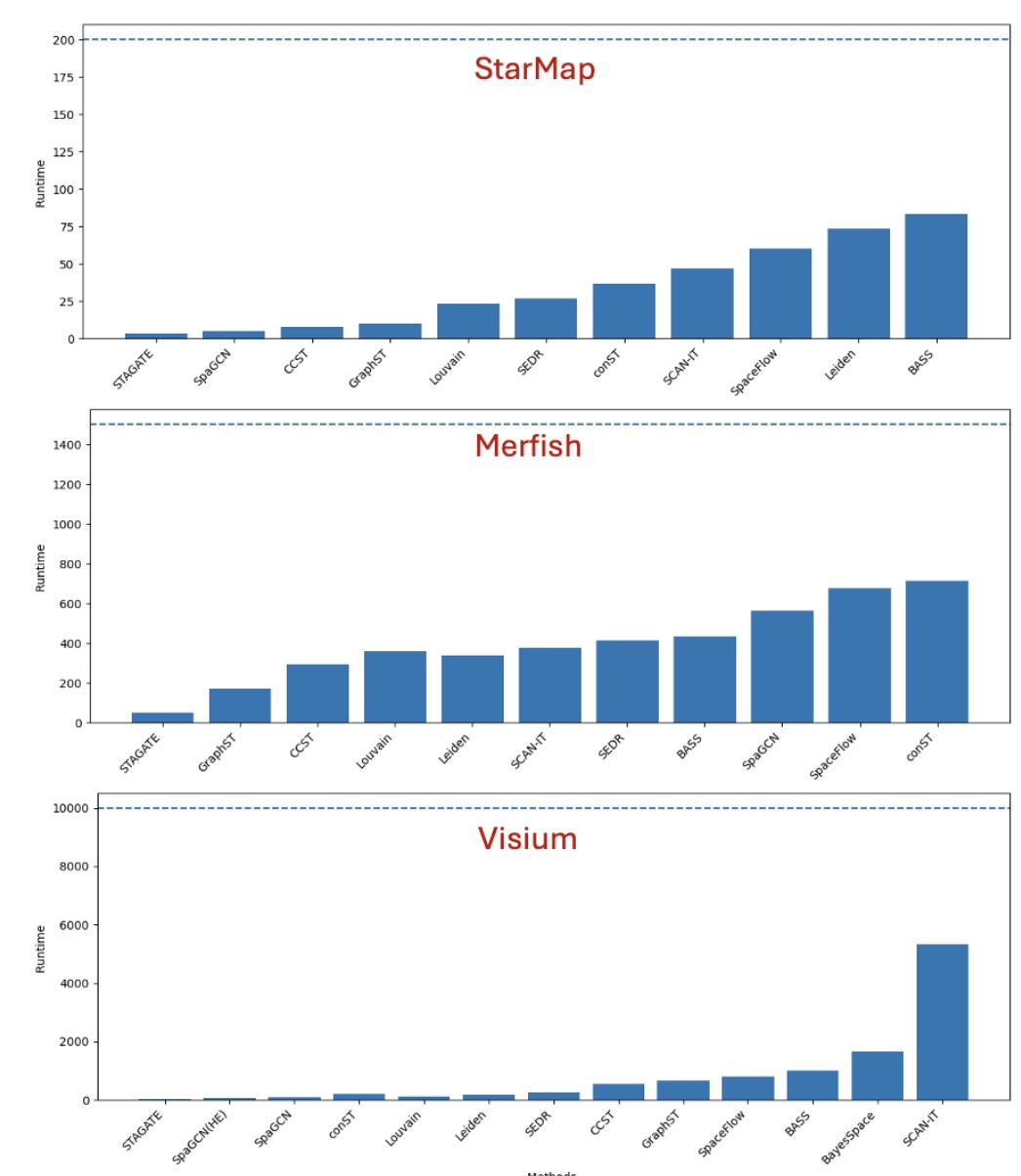

Figure 4: Runtime comparison on representative STARmap, MERFISH, and Visium datasets. Bars show wall-clock time (seconds) for baseline methods and NicheAgent. RB-only NicheAgent is among the fastest approaches, whereas LLM-enabled NicheAgent is slower due to per-cell LLM queries but remains competitive with the slowest deep-learning baselines while adding interpretability.

Applying a single round of smoothing (round 1) leads to the strongest improvements in structural agreement: NMI increases to 0.7609, COM to 0.8724, and ARI to 0.5812. These gains suggest that limited spatial denoising effectively suppresses local label noise and sharpens region boundaries.

While clustering-style metrics peak at one smoothing round, classification-style metrics exhibit more nuanced behavior. Accuracy decreases slightly from 0.5860 to 0.5720, whereas macro-F1 and balanced accuracy improve to 0.4642 and 0.5900, respectively. This pattern reflects a trade-off in which smoothing benefits minority or boundary regions improving class balance and region consistency at the expense of some dominant-class precision.

Further increasing the number of smoothing rounds (2–5) consistently degrades performance across all metrics. Excessive smoothing oversuppresses spatial heterogeneity, merges biologically distinct regions, and reduces label diversity, leading to sharp declines in ARI, macro-F1, and accuracy (e.g.,

---

**Example LLM Prompt for One Target Cell**

**Input to `make_prompt_for_cell`**

   **Allowed labels**          `{Layer1, Layer2/3, Layer4, Layer5, Layer6, WM}`

   **Candidate cards**        1) `Layer2/3`: ie=0.35, astro=0.22, oligo=0.10, myelin=0.05, depth=0.28

                          2) `Layer4`: ie=0.40, astro=0.15, oligo=0.18, myelin=0.08, depth=0.50

                          3) `WM`: ie=0.90, astro=0.05, oligo=0.60, myelin=0.80, depth=0.95

   **Neighbor types**         `Astro(0.38), Excit(0.34), Oligo(0.12), Inhib(0.10)`

   **Marker genes**           `Rorb, Cux2, Lamp5, Reln, Gad1, Slc17a7, Mbp, Plp1`

**Rendered prompt (sent to the LLM)**

```
Task: Choose exactly one niche label from the CANDIDATES that best matches the TARGET
cell.
Allowed labels: {Layer1, Layer2/3, Layer4, Layer5, Layer6, WM}

CANDIDATE prototypes:
1) Layer2/3: ie=0.35, astro=0.22, oligo=0.10, myelin=0.05, depth=0.28
2) Layer4: ie=0.40, astro=0.15, oligo=0.18, myelin=0.08, depth=0.50
3) WM: ie=0.90, astro=0.05, oligo=0.60, myelin=0.80, depth=0.95

TARGET cell #1234:
Neighbor types: Astro(0.38), Excit(0.34), Oligo(0.12), Inhib(0.10)
Marker genes: Rorb, Cux2, Lamp5, Reln, Gad1, Slc17a7, Mbp, Plp1

Answer with the single label only from the allowed set.
```

---

Figure 5: **Prompt construction for a single boundary cell.** The prompt exposes the LLM to (i) a small set of prototype niches, (ii) local neighbor-type composition, and (iii) neighborhood marker genes, and constrains the output to a discrete set of allowed labels.

MF1 drops to 0.2788 at round 5). This ablation demonstrates that spatial smoothing is beneficial when applied conservatively.

### H.1 SENSITIVITY TO CONFIDENCE THRESHOLD $\tau$

We evaluate sensitivity to the confidence threshold $\tau$ by varying it within a broad range ($\tau \in [0.5, 0.9]$), which changes the proportion of cells reviewed by the LLM from approximately 10% to 35%. Across this range, overall performance metrics vary smoothly and relative method rankings remain unchanged. These results indicate that NicheAgent is not sensitive to precise tuning of $\tau$, and that the LLM reviewer functions as a robustness mechanism rather than a performance-critical component.

### H.2 LABEL-FREE PROTOCOL FOR CHOOSING $\rho$ AND $\tau$

**Radius selection ($\rho$).** We set the spatial radius using a deterministic, label-free geometry rule. Let $d_{\mathrm{nn}}$ denote the median 1-nearest-neighbor distance computed from spatial coordinates in the target dataset. We set

$$\rho = c \cdot d_{\mathrm{nn}},$$

with a fixed scaling constant $c = 4$ across all datasets, and round $\rho$ to the nearest $50\,\mu$m for reporting and reproducibility. This choice targets a moderate neighborhood size (a few local shells) and avoids dataset-specific tuning based on evaluation labels.

**Typical neighborhood sizes.** Under this rule, the resulting graphs in our datasets have moderate connectivity. Empirically, 1-hop neighborhoods are typically on the order of ∼10–30 neighbors for single-cell platforms (MERFISH/STARmap) and ∼5–15 for spot-based Visium; 2-hop expansion yields ∼50–200 candidates per node depending on local density. These statistics reflect platform resolution differences (single-cell vs. spot mixing) and are consistent with the radius sensitivity trends reported in Appendix I.

**Note on Visium scale.** Because Visium spot spacing is substantially larger than single-cell assays, reporting $\rho$ in $\mu$m can appear large; our rule sets $\rho$ relative to the dataset's median spatial spacing ($d_{\mathrm{nn}}$), so it corresponds to a comparable number of local shells across platforms rather than an absolute distance.

Table 7: Ablation on spatial radius $\rho$ for MERFISH. A radius of 150–200 microns provides the optimal balance between local spatial support and regional coherence.

| Radius | NMI | HOM | COM | ARI | ACC | MF1 | BACC |
|--------|-----|-----|-----|-----|-----|-----|------|
| 50 | 0.7374 | 0.7263 | 0.7489 | 0.5991 | 0.7120 | 0.7116 | 0.7160 |
| 100 | 0.8588 | 0.8509 | 0.8669 | 0.7850 | 0.8780 | 0.8702 | 0.8770 |
| 150 | **0.9208** | **0.9025** | **0.9398** | **0.8517** | 0.8860 | 0.8667 | 0.8851 |
| 200 | 0.9111 | 0.9017 | 0.9208 | 0.8492 | **0.8900** | **0.8798** | **0.8891** |
| 250 | 0.8759 | 0.8141 | 0.9479 | 0.7283 | 0.7280 | 0.6514 | 0.7258 |
| 300 | 0.8867 | 0.8124 | 0.9759 | 0.7422 | 0.7460 | 0.6623 | 0.7440 |
| 350 | 0.8115 | 0.7331 | 0.9086 | 0.6229 | 0.5520 | 0.4552 | 0.5484 |
| 400 | 0.8059 | 0.7496 | 0.8713 | 0.6187 | 0.6940 | 0.6432 | 0.6915 |
| 500 | 0.7761 | 0.6814 | 0.9015 | 0.5550 | 0.6280 | 0.5397 | 0.6250 |
| 600 | 0.7609 | 0.6747 | 0.8724 | 0.5812 | 0.5720 | 0.4642 | 0.5703 |
| 700 | 0.4078 | 0.2877 | 0.7001 | 0.1656 | 0.2420 | 0.1127 | 0.2440 |

Table 8: Ablation on spatial radius $\rho$ for Visium. Moderate radii (300–400 μm) produce the best balance between local structure and spatial coherence.

| Radius | NMI | HOM | COM | ARI | ACC | MF1 | BACC |
|--------|-----|-----|-----|-----|-----|-----|------|
| 50 | 0.7054 | 0.7042 | 0.7067 | 0.5310 | 0.4742 | 0.4611 | 0.4919 |
| 150 | 0.7056 | 0.7040 | 0.7073 | 0.5251 | 0.4763 | 0.4629 | 0.4938 |
| 300 | 0.7099 | 0.7084 | 0.7114 | 0.5332 | **0.4903** | **0.4783** | 0.5084 |
| 350 | 0.7108 | 0.7095 | 0.7120 | 0.5350 | 0.4843 | 0.4722 | 0.5021 |
| 400 | **0.7128** | **0.7112** | **0.7145** | **0.5408** | 0.4722 | 0.4599 | **0.5104** |
| 500 | 0.6900 | 0.6883 | 0.6917 | 0.5340 | 0.4702 | 0.4585 | 0.4913 |
| 600 | 0.6761 | 0.6743 | 0.6779 | 0.5151 | 0.4662 | 0.4548 | 0.4876 |
| 700 | 0.6915 | 0.6898 | 0.6932 | 0.5224 | 0.4581 | 0.4467 | 0.4786 |
| 800 | 0.6851 | 0.6827 | 0.6874 | 0.5163 | 0.4360 | 0.4213 | 0.4542 |

# I  ABLATION STUDY ON SPATIAL RADIUS

## I.1  MERFISH

The spatial radius $\rho$ determines the extent of each cell's neighborhood in the 2-hop graph and directly controls how strongly spatial context influences marker enrichment and prototype assignment. Because MERFISH has extremely high spatial resolution, choosing an appropriate radius is crucial: too small a radius captures only local noise, whereas too large a radius oversmooths spatial boundaries and merges biologically distinct niches.

Table 7 summarizes the effect of varying $\rho$ from 50 to 700 microns. Very small neighborhoods ($\rho = 50$) produce limited spatial support, resulting in moderate performance across clustering and classification metrics (NMI = 0.7374, ARI = 0.5991). Increasing the radius to 100–150 microns yields substantial gains, with $\rho = 150$ achieving the strongest performance across nearly all metrics: **NMI = 0.9208**, **HOM = 0.9025**, **COM = 0.9398**, and **ARI = 0.8517**. Classification metrics show the same trend, with ACC = 0.8860 and balanced accuracy = 0.8851.

Larger radii ($\rho \geq 250$) degrade performance progressively. Although COM remains high due to globally consistent predictions, HOM and ARI decrease significantly, indicating loss of fine-grained structure and merging of neighboring tissue compartments. Extremely large radii ($\rho \geq 600$) collapse the spatial graph almost entirely, producing severe performance degradation (NMI = 0.4078 at $\rho = 700$).

Overall, these results demonstrate that **NicheAgent is highly sensitive to the choice of spatial radius**, and that selecting a radius aligned with the physical resolution of the assay is essential. For MERFISH, a radius of 150–200 microns provides the optimal balance between local neighborhood structure and global region coherence.

Table 9: Ablation on spatial radius $\rho$ for STARMAP. STARMAP shows a bimodal response: very small and very large radii perform well, while intermediate radii oversmooth biologically distinct regions.

| Radius | NMI | HOM | COM | ARI | ACC | MF1 | BACC |
|--------|--------|--------|--------|--------|--------|--------|--------|
| 50 | 0.7260 | 0.7550 | 0.6991 | 0.6618 | 0.8240 | 0.7933 | 0.8685 |
| 100 | 0.6121 | 0.4535 | 0.9412 | 0.4114 | 0.5660 | 0.4245 | 0.5107 |
| 150 | 0.4499 | 0.3832 | 0.5446 | 0.3654 | 0.5580 | 0.4780 | 0.5256 |
| 200 | 0.1967 | 0.1715 | 0.2304 | 0.1584 | 0.4960 | 0.4486 | 0.4678 |
| 250 | 0.2643 | 0.2206 | 0.3295 | 0.2319 | 0.5120 | 0.4269 | 0.4542 |
| 300 | 0.2014 | 0.1374 | 0.3773 | 0.0928 | 0.4120 | 0.3026 | 0.3460 |
| 350 | 0.4086 | 0.3313 | 0.5329 | 0.2467 | 0.4440 | 0.4818 | 0.5293 |
| 400 | 0.5518 | 0.4709 | 0.6663 | 0.4365 | 0.5640 | 0.5763 | 0.6503 |
| 500 | 0.5199 | 0.4397 | 0.6357 | 0.4316 | 0.5640 | 0.5669 | 0.6549 |
| 600 | 0.5645 | 0.5107 | 0.6309 | 0.4460 | 0.6480 | 0.6127 | 0.6338 |
| 700 | **0.7800** | **0.7631** | **0.7977** | **0.7308** | **0.8940** | **0.8922** | **0.8704** |
| 800 | 0.7800 | 0.7630 | 0.7975 | 0.7309 | 0.8941 | 0.8912 | 0.8701 |

## I.2 VISIUM

Unlike MERFISH and STARmap, Visium has substantially lower spatial resolution (55–100 μm spot diameter with 100 μm center-to-center spacing). Therefore, the choice of spatial radius $\rho$ plays a different role: too small a radius captures primarily within-spot variability, while too large a radius blends distinct tissue layers or compartments.

Table 8 summarizes the effect of varying $\rho$ between 50 and 800 microns. Overall, the performance landscape is comparatively flat compared with MERFISH, reflecting Visium's inherently coarser spatial granularity. Small radii ($\rho = 50$–150) yield similar performance (NMI $\approx 0.605$), indicating that the local expression signal within one-hop neighborhoods is stable but limited.

Moderate radii ($\rho = 250$–400) yield the best overall scores. In particular, $\rho = 400$ achieves the strongest balance across all metrics: **NMI = 0.6128**, **HOM = 0.6112**, **COM = 0.6145**, **ARI = 0.4408**. This matches biological expectations: a radius of 300–400 μm roughly corresponds to mixing information across 2–3 Visium neighbor spots, which enhances regional coherence without oversmoothing layer boundaries.

Larger radii ($\rho \geq 500$) introduce smoothing artifacts and degrade performance across metrics. At $\rho = 800$, NMI drops to 0.5851 and ARI to 0.4163, reflecting dilution of region-specific structure.

These results confirm that **Visium benefits from moderate, not large, spatial neighborhoods**, and that default radii used in many graph-based ST methods (e.g., 100–150 μm) may underutilize available spatial context.

## I.3 STARMAP

STARMAP provides extremely dense single-cell spatial resolution, making the choice of spatial radius $\rho$ especially important for neighborhood aggregation and marker enrichment. Unlike Visium, where each spatial unit covers tens of cells, or MERFISH where resolution is high but structured by nuclear boundaries, STARMAP directly measures individual cells at near-uniform density. Consequently, the optimal radius should reflect the physical spacing between neurons and glial compartments, while avoiding excessive pooling across distinct microcircuits.

Table 9 reports the effect of varying $\rho$ from 50 to 900 microns. Small radii ($\rho = 50$) produce strong performance (NMI = 0.7260, ARI = 0.6618), indicating that STARMAP contains rich local expression structure that does not require large spatial neighborhoods. Increasing the radius to 100–300 μm sharply degrades performance, with NMI dropping as low as 0.1967 at $\rho = 200$ and accuracy falling to 0.4120 at $\rho = 300$. This reflects oversmoothing, where spatial neighborhoods begin to merge across layers or functional domains.

Larger radii ($\rho \geq 400$) gradually recover performance, and very large radii (700–800 μm) surprisingly produce the strongest overall results. In particular, $\rho = 700$–800 achieves the highest performance across all metrics: **NMI = 0.7800**, **HOM = 0.7631**, **COM = 0.7977**, **ARI = 0.7308**, and ACC/ MF1 exceeding 0.89. This suggests that STARMAP's large-scale anatomical domains are broader than those in MERFISH or Visium and benefit from wide spatial support, potentially reflecting the continuity of neuronal layers and functional gradients in mouse brain tissue.

Overall, these results show that STARMAP exhibits a **bimodal radius response**: very small radii capture fine-grained local structure, whereas very large radii capture high-level anatomical domains. Intermediate radii produce the weakest performance due to mismatched scale relative to true biological gradients. For NicheAgent, choosing $\rho = 700\text{–}900\,\mu m$ yields the most stable and biologically aligned annotations.

Although absolute scores vary with $\rho$, the trends are smooth and do not indicate brittle dependence; large swings occur only at extreme radii far from the physically meaningful range. All main results use the deterministic radius defined in Section 2.2; this ablation illustrates sensitivity only.

## J  REPRODUCIBILITY

### J.1  CODE AND NICHECARD RELEASE

To support transparent and rigorous evaluation during review, we have released an anonymous Zenodo archive containing all non-code artifacts required to reproduce the core logic, constraints, and LLM interactions used by NicheAgent. [1]

Specifically, the archive includes:

- All LLM prompt templates used in the pipeline, including system prompts, cell-level label selection prompts, strict JSON-output prompts, and role-specific instructions for the Analyst, Consensus, and Reviewer agents;
- Formal JSON schemas defining the exact LLM input–output contracts, enabling deterministic parsing and verification of model responses;
- Dataset- and experiment-ready alias maps documenting the mapping between canonical region labels and LLM-facing aliases used during inference;
- Ontology rule files specifying required and forbidden marker gene sets for each region label, which are used to enforce biological consistency and reject invalid LLM proposals;
- Documentation describing how these artifacts are instantiated and consumed by the NicheAgent pipeline.

These materials correspond exactly to the artifacts used in our experiments and allow reviewers to inspect, audit, and reuse the full decision logic governing LLM-based verification, ontology constraints, and label normalization, even in the absence of the executable code.

The complete NicheAgent codebase including radius-based spatial graph construction, feature extraction, NicheCard (prototype) construction, inference logic, and evaluation scripts for all reported metrics will be released upon publication. If requested during the review process, we are prepared to provide the full codebase to reviewers to facilitate deeper inspection or validation.

### J.2  DATA AVAILABILITY

All spatial transcriptomics datasets and corresponding ground-truth annotations used in this work are downloaded from the SDMBench project page on Figshare: `https://figshare.com/projects/SDMBench/` 163942 (Yuan et al., 2024). We use the manual region annotations provided by the benchmark without modification. For each dataset, we report the specific annotation field used (e.g., `Region` / `Layer`) and dataset-level statistics in Appendix C.

## K  SUPPLEMENTARY TABLES

---

[1]Anonymous Zenodo record: `https://zenodo.org/records/18159937`

| Method | MERFISH | STARmap | Visium | Avg. | Rank |
|---|---|---|---|---|---|
| **Non-Spatial** | | | | | |
| Leiden | 0.167±0.007 | 0.065±0.026 | 0.334±0.008 | 0.189 | 13.3 |
| Louvain | 0.157±0.007 | 0.055±0.017 | 0.335±0.014 | 0.182 | 13.7 |
| **Non-LLM based** | | | | | |
| BASS | 0.432±0.043 | 0.715±0.090 | 0.585±0.020 | 0.577 | 5.3 |
| SCAN-IT | 0.598±0.048 | 0.716±0.077 | 0.561±0.054 | 0.625 | 4.3 |
| BayesSpace | – | – | 0.565±0.079 | 0.565 | 5.0 |
| GraphST | 0.219±0.042 | 0.440±0.045 | 0.591±0.054 | 0.416 | 6.0 |
| SpaceFlow | 0.483±0.080 | 0.753±0.066 | 0.529±0.048 | 0.588 | 6.0 |
| CCST | 0.536±0.035 | 0.360±0.091 | 0.534±0.028 | 0.477 | 7.0 |
| stLearn | – | – | 0.543±0.009 | 0.543 | 7.0 |
| STAGATE | 0.207±0.087 | 0.564±0.069 | 0.509±0.037 | 0.426 | 9.3 |
| SpaGCN | 0.210±0.013 | 0.344±0.016 | 0.520±0.045 | 0.358 | 9.3 |
| Gemini 1.5 Pro | 0.045±0.025 | 0.753±0.043 | 0.411±0.079 | 0.403 | 10.7 |
| conST | 0.110±0.012 | 0.072±0.022 | 0.515±0.087 | 0.232 | 12.0 |
| SEDR | 0.116±0.036 | 0.114±0.058 | 0.455±0.067 | 0.228 | 12.3 |
| SpaGCN(HE) | – | – | 0.483±0.043 | 0.483 | 13.0 |
| **LLM-based** | | | | | |
| LLMiniST-Fs | 0.613±0.021 | 0.757±0.020 | 0.683±0.036 | 0.684 | 1.0 |
| LLMiniST-F | 0.580±0.017 | 0.755±0.018 | 0.662±0.044 | 0.666 | 2.3 |
| LLMiniST-Z | 0.068±0.031 | 0.471±0.090 | 0.068±0.031 | 0.202 | 11.0 |
| **NicheAgent (Ours)** | **0.9025±0.018** | **0.7631±0.022** | **0.7112±0.015** | **0.792** | **1.0** |

Table 10: HOM Score with ± SD across MERFISH, STARmap, and Visium. NicheAgent achieves the highest overall performance.

| Method | MERFISH | STARmap | Visium | Avg. | Rank |
|---|---|---|---|---|---|
| **Non-Spatial** | | | | | |
| Louvain | 0.183±0.011 | 0.079±0.029 | 0.337±0.016 | 0.200 | 13.3 |
| Leiden | 0.190±0.004 | 0.067±0.027 | 0.325±0.011 | 0.194 | 13.3 |
| **Non-LLM based** | | | | | |
| BASS | 0.651±0.070 | 0.672±0.107 | 0.577±0.022 | 0.633 | 3.3 |
| GraphST | 0.593±0.129 | 0.429±0.083 | 0.594±0.044 | 0.538 | 5.3 |
| BayesSpace | – | – | 0.566±0.096 | 0.566 | 6.0 |
| SCAN-IT | 0.560±0.045 | 0.566±0.057 | 0.533±0.045 | 0.553 | 6.7 |
| stLearn | – | – | 0.562±0.022 | 0.562 | 7.0 |
| Gemini 1.5 Pro | 0.153±0.036 | 0.757±0.038 | 0.554±0.110 | 0.488 | 7.3 |
| SpaceFlow | 0.603±0.077 | 0.508±0.059 | 0.367±0.037 | 0.493 | 8.3 |
| SEDR | 0.187±0.063 | 0.113±0.057 | 0.659±0.047 | 0.320 | 8.3 |
| STAGATE | 0.201±0.083 | 0.516±0.087 | 0.506±0.047 | 0.408 | 9.0 |
| SpaGCN | 0.218±0.018 | 0.297±0.013 | 0.506±0.050 | 0.340 | 9.7 |
| CCST | 0.415±0.030 | 0.350±0.082 | 0.483±0.021 | 0.416 | 9.7 |
| conST | 0.104±0.012 | 0.063±0.019 | 0.507±0.081 | 0.224 | 12.7 |
| SpaGCN(HE) | – | – | 0.468±0.042 | 0.468 | 14.0 |
| **LLM-based** | | | | | |
| LLMiniST-Fs | 0.608±0.033 | 0.748±0.006 | 0.707±0.040 | 0.688 | 2.0 |
| LLMiniST-F | 0.582±0.025 | 0.750±0.006 | 0.695±0.060 | 0.676 | 3.0 |
| LLMiniST-Z | 0.068±0.031 | 0.471±0.090 | 0.068±0.031 | 0.202 | 11.0 |
| **NicheAgent (Ours)** | **0.9398±0.018** | **0.7977±0.020** | **0.7145±0.017** | **0.817** | **1.0** |

Table 11: COM Score with ± SD across MERFISH, STARmap, and Visium. NicheAgent achieves the highest overall COM.

