# OpenReview forum: "CONSTRAINED LANGUAGE-GUIDED REFINEMENT FOR ZERO-SHOT SPATIAL ANNOTATION"
_ICLR.cc/2026/Workshop/LMRL — Submitted to ICLR 2026 Workshop LMRL_

### Official Review · Reviewer_pfRW · 2026-02-17
**Interesting idea, but it lacks a robust evaluation and may have potential data leakage.**

**Rating:** 3
**Confidence:** 4

**Review:**

The authors present NicheAgent, a zero-shot training-free framework to identify spatial niches for spatial transcriptomics. NicheAgent first uses Nichecards, region prototypes derived from external datasets, to assign cells to their nearest prototype. Low-confidence assignments then get reviewed by an LLM. Finally, the assignments get spatially smoothed with majority voting.

Strength:
- Diverse experiments: tested on 3 platforms (STARmap, MERFISH, Visium) and compared against 17 other methods
- Ablation studies are present and show the impact of the LLM reviewer, the number of spatial smoothing rounds, and the sensitivity of spatial radius for neighborhood/impact of local neighborhood.

Weakness:
- Missing details on how the data is preprocessed and hyperparameters? Is it identical across methods?
- Missing downstream analyses with the niche assignments (e.g., spatial_scatter, pathway analysis) to show biological utility.
- Dataset information misses tissue types (e.g., Visium: LIBD Human Brain dataset with samples from the DLPFC region). It determines the importance e.g. of spatial information (e.g,. structured tissue) and might also correlate to the results in the ablation studies (e.g. spatial radius for neighborhood).
- The results section didn’t link results to the biological characteristics of each dataset.
- Why only 2-hop neighborhood expansion?
- ρ and τ seem to be optimised and selected on all datasets (missing train and test split); undermining the "label-free generalization" claim
- NicheCards are derived from SDMBench datasets, the same collection used for evaluation. An explicit subject-level separation table is required to verify no overlap
- Gene panel intersection sizes across platforms are never reported
- All three evaluated datasets are coming from brain tissue
- Unclear how the authors matched the number of clusters (k) across different methods (e.g., resolution-based Leiden vs. number-based GraphST vs. NicheAgent) in Table 1? This is important for a fair comparison of NMI and ARI.
- The "training-free / zero-shot" framing is misleading. Prototype centroids computed from annotated reference data encode fully supervised signals. The correct term is reference-guided annotation, and natural baselines (Seurat label transfer, Symphony) are absent

Critical comments:
-  NicheAgent seems to be missing in the runtime comparison in Figure 4 (Appendix).

---

### Official Review · Reviewer_SaM3 · 2026-02-25
**The quality of presentation and the lack of discussion prevent the full appreciation of interesting results**

**Rating:** 5
**Confidence:** 3

**Review:**

**Quality**

The paper presents a method that appears sound and well adapted to the problem. However, the paper is held back by the lack of clarity in its presentation as detailed below. Although the core contribution is supported by a strong performance against the baselines, the role of the LLM component and the choice of the prototypes would benefit from a better contextualisation.

**Clarity**

The quality of presentation is inconsistent. While the motivation and the methodological part is rather well presented, the paper becomes significantly more difficult to follow as it gets to the evaluation section. Details are missing from the Appendix (even when referenced). The navigation between the Appendices and their sequencing is often unclear.

The authors might consider modifying the references to the Appendix to explicitly mention subsections (e.g. additional details in H.2).

The labelling convention is not fully explained:
* Line 372: *...and Balanced Accuracy (BACC) when ground truth labels are available...* It is unclear how many cells/ spots have missing or mismatched annotations w.r.t. the annotations of the prototypes.

Similarly a high-level overview of the labels used, their interpretation, and correspondence should be provided (or possibly referenced) for a wider audience.

Examples below could benefit from further clarification:

* missing definitions: (Section 2.8) silhouette score in the 50-dimensional PCA space.

* missing definitions: (Page 15) HVGs, the number and selection of which is unclear. Needs a separate introduction in the data part or additional details in F.6.

* Broken reference: (line 1077) Table 3.

* Possible inconsciences and redundancies: (Appendix B) **Prototype distances and feature space alignment**. mentions library normalisation of the target dataset gene counts. The following **Cross-Platform Gene Harmonization** additionally mentions the log-norm but not the lib-norm. Then the prototype dataset gene preprocessing is further discussed in C.1, C.2, and F.6. The Comparability of feature scales between target and prototypes for distance computation is mentioned several times but I am not sure which normalisation was chosen in the end.

Minor remarks and optional suggestions:

* missing definitions, redundancy: (Section 2.2) the definition of $\rho$ is possibly redundant  (The paragraph on the deterministic radius selection and label-free selection), \tau is not clearly introduced. Appendix B: functions like `build_nichecards` are ambiguous.  It is a minor issue assuming they refer to the code that will be released.

* formatting and broken references: (Page 19) the `\paragraph`s on numerical stability and robustness can separated from the metrics with a subsection for readability. Appendix X (line 1012) is not defined.

* Line 226 possibly starts with a missing \paragraph

* structure: Appendices H.1 and H.2 do not appear to belong to the Appendix H conceptually.

Overall, a single appendix section that clearly defines the dataset and the preprocessing pipelines used could be beneficial (some parts of B, C, and F extracted and re-combined together).

**Originality and significance**

The paper references a diverse selection of prior works on ontology-constrained LLM inference and annotation for spatial transcriptomics data. Individual steps of the algorithm rely on previously explored ideas, yet the overall method and its framing are original and bring considerable improvements.

While the performance gap from the baselines is impressive, two considerations complicate the interpretation of the results' significance. First, the reported baselines are relatively weak. I would like to know the author's interpretation of the results reported in Table 1, notably how much of this gap can be attributed to hyperparameter optimisation and poor domain adaptation of other methods. Second, the rule-based model is already a strong performer, which raises the question of the contribution of sensitivity to relevant prototype selection.

Nonetheless, the Rule-based model appears to be a robust and significant improvement over the baseline methods with extra performance boost provided by the LLM component.

**Pros**

* [P1] The method and its framing. The authors emphasise the reviewer/refiner role of the LLM and put a lot of emphasis on using them to complement reasonable annotation heuristics.

* [P2] The strong rule-based ablated baseline is a solid foundation for future improvements beyond domain-agnostic LLMs.

* [P3] The authors ensure that the prototype-generating and the target sets are independent.

* [P4] Careful handling of the heuristics. The analysis of sensitivity to $\rho$, to the number of hops, and to spatial smoothing across datasets helps better understand the role of geometric organisation of the measurements and the limitations of the method.

**Cons**

* [C1] Inconsistent presentation as discussed above.

* [C2] Missing details. a) Datasets: the number of samples used for prototype creation and the biological priors involved in their selection (source, tissue, HVG selection after intersection etc.); the number of samples with missing annotations the extent of mismatches between the external prototypes and the ground-truth annotations. b) Evaluation: a more in-depth analysis for sensitivity w.r.t. $\tau$ in H1 (conducted per dataset, and including extreme value of $\tau$ similarly to the provided analysis for $\rho$).

* [C3] The ablated Rule-Based-only model is not compared to all the other baselines on the same clustering metrics.

* [C4] Given the reported overperformance w.r.t. the baselines on MERFISH and Visium a more in-depth discussion of the results and limitations could be expected.

**Questions**
The impact of data used for the prototypes

* (Prototype dataset cell, subject, replicate counts). Can the authors provide additional details and statistics regarding the dataset used in prototype definition (across Visium, STARmap, and MERFISH)?

* Can the authors provide clearer definitions of regions $\mathcal{R}$ as well as a distribution of samples for which the ground-truth annotation is available/ matches the annotations of the prototypes.

* How would the authors evaluate transferability of the constructed NicheCards to OOD samples.

* What are the main limitations of the proposed methodology considered by the authors?

* (not impacting the evaluation) Have the authors considered other methods of label refinement and/or domain specific LLMs as baselines for NicheAgent?

---

### Official Review · Reviewer_basG · 2026-02-26
**Well-written but concerns about reference data used**

**Rating:** 5
**Confidence:** 4

**Review:**

Summary: The authors propose an LLM-based method for spatial annotation that combines reference dataset-derived priors with an LLM reviewer.

Strengths:
 - The paper is well-written. There is a clear scope, description, and design pattern.
 - The paper clearly builds upon previous benchmarks and LLM-based methods.
 - The metrics used are comprehensive and easy to compare to popular benchmarks like SDMBench.

Weaknesses:

- While the LLM-based method performs best, this seems very heavily reliant on “nichecards” and appropriate external reference data. In addition, the performance gain in Table 4 from adding the LLM reviewer is not large. This is a significant part of the work and needs more explanation.
 - There should be more clarity on how many datasets and which datasets exactly were used to compute the nichecards.
- There should also be more biological description of the datasets. How much external data is needed, and how similar does the external data need to be to generalize well? What tissues were used? This will clarify how useful the method is in a general context.
- The SDMBench benchmark makes a distinction between “data” and “datasets,” where different “data” can refer to different slices of the same tissue, and different “datasets” are different experiments/technologies. Does this work ensure that this separation is done?
- The comparison between the proposed method and most other methods is somewhat misleading since those methods do not use external datasets. The task of unsupervised clustering/post-hoc cluster labeling (most other methods) vs. assigning cells to known domain labels (this method) is quite different.

---

### Meta-Review · Area_Chair_5h4C · 2026-02-25

**Recommendation:** Reject
**Confidence:** 4

**Metareview:**

I recommend rejection.

---

### Decision · Program_Chairs · 2026-03-02

**Decision:**

Reject

**Comment:**

Please see the meta-review.